# Effect of crocin and treadmill exercise on oxidative stress and heart damage in diabetic rats

**Laleh Pourmousavi, Rasoul Hashemkandi Asadi** [ORCID]**\*, Farzad Zehsaz, Roghayeh Pouzesh Jadidi\***

Department of Physical Education and Sport Sciences, Tabriz Branch, Islamic Azad University, Tabriz, Iran

\* hashemkandirasoul@gmail.com

## Abstract

Diabetes increases the production of free radicals and inflammatory agents in the heart tissue and alters the expression of genes associated with the induction of apoptosis. Considering the importance of common cardiovascular disorders in diabetes, this study investigated the effect of eight weeks of aerobic exercise and crocin use, as well as tissue damage and oxidative stress caused by diabetes in the hearts of adult rats. Streptozotocin 50 mg/kg was injected as a single dose intraperitoneally to cause the diabetes. After 72 hours, a glucometer monitored blood glucose levels, and blood glucose above 250 mg/dl was considered diabetes. Continuous treadmill exercise was performed for eight weeks by placing the animal on the treadmill. Next, the animals were anesthetized, and samples were taken from the hearts and frozen in liquid nitrogen. Then, superoxide dismutase (SOD), glutathione peroxidase (GPx), and malondialdehyde (MDA) were measured in the cardiac tissue. Finally, the hearts of half of the animals were immediately immersed in a formalin solution for histological changes. According to our findings, diabetes increased lipid peroxidation, characterized by increased MDA levels in the control diabetes group and decreased SOD and GPx levels ($P$<0.05). It also changes the balance of expression of genes associated with apoptosis control, increased Bcl-2-associated X (Bax) expression, and decreased Bcl-2 expression ($P$<0.05). Also, we observed the induction of apoptosis in cardiac tissue. Using eight weeks of continuous exercise and administration of crocin significantly reduced blood sugar levels and lipid peroxidation and increased the activity of antioxidant enzymes and Bcl-2 gene expression compared to the diabetes control group. In addition, continuous exercise and crocin improved the oxidative stress parameters in the control group. This study showed that diabetes could cause oxidative stress and heart dysfunction. Moreover, simultaneously and separately, aerobic exercise with a treadmill and crocin administration can reduce these disorders and prevent apoptosis in the heart tissue.

**Data Availability Statement:** We have several legal and official restrictions for the general and public availability of datasets used and analyzed in the current study, including: Due to ethics committee

and funding organizations policies by Tabriz University of Medical Sciences and Islamic Azad, the University of Tabriz restrictions do not agree to share datasets publicly. Any access to datasets should be requested reasonably. We present contact information (Phone, E-mail, and Address) for free authors when they need to access data: https://ethics.research.ac.ir/PortalCommittee.php?code=IR.IAU.TABRIZ.REC https://tbzms.iau.ir/fa/contactus

**Funding:** The author(s) received no specific funding for this work.

**Competing interests:** The authors have declared that no competing interests exist.

## 1. Introduction

Diabetes or diabetes mellitus (DM) is a chronic endocrine disease with widespread concern worldwide. It is a heterogeneous metabolic disorder caused by a lack of insulin production in the body or insulin resistance that can impair heart function. Diabetes increases the apoptosis rate (Pancreatic β-cell apoptosis is also a pathological feature common to Type 1 diabetes mellitus (T1DM) and T2DM. In T2DM, insulin resistance with visceral obesity leads to a glucose toxicity effect, which accelerates β-cell death by apoptosis) in heart cells and disrupts cardiac function [1]. In many diabetic patients, defects in cardiac activity are observed in the form of decreased blood flow, heart rate, and atherosclerosis [2]. Diabetes mellitus is a syndrome characterized by lipid, carbohydrate, and protein metabolism disruption due to elevated blood sugar levels. Consequently, it has the potential to heighten the susceptibility to vascular disease [3].

Although the mechanism of DM is not well understood, the primary damaging agents include increased free radicals and oxidative stress [4]. The presence of antioxidants such as vitamins or flavonoids in the diet can have protective effects in diabetic patients [5]. Excessive production of reactive oxygen species (ROS) can cause damage to the mitochondrial membrane and cytochrome C release process, resulting in apoptosis in heart tissue cells [6]. Apoptosis, also known as programmed cell death, is an inherent and dynamic biological process that occurs in response to exposure of cells to cytotoxic substances, as observed throughout evolution. Apoptosis is characterized by several prominent aspects, namely cellular shrinkage, membrane impairment, chromatin condensation, and fragmentation of DNA. Multiple proteins have a role in the regulation of apoptosis. The Bcl-2 protein family encompasses a group of proteins that exert regulatory control over apoptosis, playing pivotal roles in inhibiting and promoting this cellular process. While the Bcl-2 protein functions as a suppressor, the Bax protein serves as a promoter of apoptosis [7].

Crocin and crustin are the predominant carotenoids that play a pivotal role in determining the coloration of saffron. Crocin undergoes metabolic processes within the human body, forming crustin, which exhibits numerous medicinal qualities. It is a potent antioxidant and anti-inflammatory agent in laboratory animals; it can increase the efficacy of antibiotics, reduce cholesterol, and reduce the inhibitory effect on carcinogenicity. About the suppressive impact of crocin on tumorigenesis, a plausible hypothesis may be formulated suggesting that crocin functions as a stimulator for a DNA-repair enzyme that has not yet encountered DNA. However, other mechanisms have been reported for their protective effect on DNA, including its direct interaction [8, 9].

Tissues exposed to increased oxidative stress for a long time are adapted to the antioxidant system by stimulating the enzymatic activity of superoxide dismutase (SOD), which involves increasing the action of the enzyme in rats. It has been shown that oxidative stress induced by acute exercise can affect the erythrocytes of non-exercised rats" and "exercised rats, while it does not significantly affect the erythrocytes of exercised rats. However, mild to moderate exercise can reduce oxidative stress by increasing the regular excretion of antioxidants and reducing the complications of diabetes [9]. Different studies evaluated the effect of crocin on cardiovascular damage [10–12]. Accordingly, this study aimed to determine the effect of forced treadmill exercise and concomitant treatment with crocin on heart tissue damage and oxidative stress in diabetic rats.

## 2. Materials and methods

### Animals

This experimental study included 64 male Wistar rats weighing 200–250 g. The animals were prepared from the animal house of Tabriz Medical School and kept in standard conditions (temperature 22–24°C with a 12-hour light/dark cycle). All animal procedures were according to the guidelines by the ethics committee of the Islamic Azad University of Tabriz, Iran, in 2021.

To induce type 1 diabetes, streptozotocin (STZ) was injected via the tail vein at a dose of 50 mg/kg in 0.1 mol/L citrate buffer (pH 4.5) or citrate buffer alone as a control under anesthesia [13]. One week after the STZ injection, rats exhibiting hyperglycemia were considered type 1 diabetic and subjected to subsequent experiments. After the completion of experiments, rats were deeply anesthetized with sodium pentobarbital (60 mg/kg), and hearts were rapidly excised and frozen in liquid nitrogen for later analysis.

### Animal preparation and study design

After being kept in standard conditions for two weeks, the animals (64) were randomly divided into eight equal groups (n = 8) as follows: Group 1: Healthy group receiving saline by gavage; Group 2: Diabetic group receiving saline by gavage; Group 3: Diabetic group receiving crocin (50 mg/kg) by gavage; Group 4: Healthy group receiving crocin (50 mg/kg) by gavage; Group 5: Healthy group exercising on a treadmill as planned; Group 6: Healthy group exercising on a treadmill and receiving crocin (50 mg/kg); Group 7: Diabetic group exercising on a treadmill and receiving crocin (50 mg/kg) by gavage; and Group 8: Diabetic group exercising regularly.

### Induction of diabetes and treatment

First, we measured the glucose levels of rats in all the intervention and control groups. Then, streptozocin was injected intraperitoneally (50 mg/kg) [14]. After 72 hours, blood glucose levels were measured by an EASY GLOCO once daily. Then, crocin at a 50 mg/kg dose by gavage to the rats.

### Exercise protocol

According to previous studies, if we do not run the maximum effort test ($VO_2$ max test for diabetic rats) after the rats are adapted to run on the treadmill and in the first test of the maximum running speed of the rat ($VO_2$ max), the maximum speed of the diabetic rat is 19.58 m/min (40–60% $VO_2$ max equivalent to a rate of 11.75–7.83 m/min). Five weeks after moderate-intensity continuous aerobic exercise training (MICT), the maximum running speed of the rat ($VO_2$ max) will be equal to 25.83 m/min (40–60% $VO_2$ max, similar to the rate of 15.83–5.55 m/min). Therefore, based on previous studies, we implemented MICT (equal to 40–60% of the maximum running speed of rats ($VO_2$ max)) for eight weeks in two stages to adapt and apply the standard training overload.

In the first stage (first to fourth weeks), the rats ran on the treadmill for 60 minutes with a speed of 11.75–7.83 m/min (equivalent to 40–60% $VO_2$ max), and in the second stage (fifth to eighth weeks) they ran with a speed of 10/55 m/min (equivalent to 40–60% of $VO_2$ max) five sessions per week. To achieve adaptation to the exercises, we increased the intensity and duration of the training. Each training session consisted of three steps: 1) warm-up (5 minutes with 30% $VO_2$ max), 2) MICT (for 60 minutes with 40–60% $VO_2$ max), and 3) return to the original state (5 minutes with 30% $VO_2$ max) [15].

## Serum glucose, HDL-C and LDL-C assays, and cholesterol levels

At the beginning of the study, blood glucose levels were determined using a portable glucometer on samples collected from the tip of the tail vein. At the end of the study, serum glucose levels were determined using commercial kits (Parsazmun, Iran). The value was expressed in mg/dl. Serum insulin levels were measured by enzyme-linked immunosorbent assay (ELISA) using the commercial kit of Rat Insulin (Mercodia). The serum hormone levels were determined using an ELISA kit (Demeditec Diagnostics, Germany).

The serum levels of urea and creatinine were assayed using the method of the Parsazmun kit. Plasma malondialdehyde (MDA) levels were measured in the following way. First, 0.20 ml of serum was added to a microtube containing 3 ml of glacial acetic acid, then 1% barbituric acid (TBA in 2% NaOH) was added to the microtube. The tube was then placed in boiling water for 15 minutes. After cooling, the adsorption of the resulting solution was read as pink at 532 nm [16, 17]. The concentrations of and glutathione peroxidase (GPx) in the heart tissue were assessed using an ELISA reader (Antus) according to the manufacturer's protocols (Ransod and Randox UK).

## Histopathological evaluation

The rat heart tissue sample was fixed in a formalin 10% solution for 48 h and embedded in paraffin blocks. The blocks were then sectioned at 5 μm, deparaffinized, and stained with hematoxylin and eosin (H&E). Three slides were taken from each heart tissue's upper, lower, and middle parts. Afterward, they were thoroughly scrutinized by a light microscope (Nikon) by two independent judges in a blind condition [18, 19].

## Biochemical analysis

The tissue-oxidative stress markers were measured based on our previous study [20]. Also, the protein concentration of the heart tissues was determined using the Bradford method [21]. Briefly, samples were homogenized in a solution of potassium chloride (5.1%) to measure the MDA levels of heart tissues to obtain 1:10 (w/v) whole homogenates. Then, using the thiobarbituric-acid reaction, the levels of MDA in homogenized tissues were calculated according to the method of Uchiama and Mihara [22]. Besides, the levels of SOD and GPx were measured according to the techniques developed by Srivivasan *et al.* [23] and Paglia and Valentine [24], respectively, using commercial kits (Ransod and Ransel, Randox Com, UK).

## Quantitative real-time polymerase chain reaction (RT-qPCR)

On the last day of the study, to analyze the mRNA expression of Bax and Bcl-2 in heart tissues of rats, the total RNA of heart tissues was isolated using a TRIzol reagent kit (Invitrogen, Paisley, UK), based on our previous study [18]. DNase I treatment was performed after RNA extraction using the RNeasy Micro kit (Invitrogen Life Technologies, Carlsbad, CA, USA) to

**Table 1. mRNA primers for real time PCR.**

| Genes | Primer sequence (5′→3′) |
|---|---|
| **Bax** | Forward: GGCGAATTGGAGATGAACTG<br>Reverse: TTCTTCCAGATGGTGAGCGA |
| **Bcl-2** | Forward: CTTTGCAGAGATGTCCAGTCAG<br>Reverse: GAACTCAAAGAAGGCCACAATC |
| **GAPDH** | Forward: GGCACAGTCAAGGCTGAGAATG<br>Reverse: ATGGTGGTGAAGACGCCAGTA |

eliminate genomic DNA contamination. Then, after determining the RNA concentration of samples at the absorbance of 260 and 280 nm, the obtained RNA was eluted with RNase-free water and adjusted to a concentration of 0.7 μg/ml. Next, complementary DNA (cDNA) was generated in a total volume of 20 μl using the commercial kit (Thermo Scientific, EU). The reaction of RT-qPCR was conducted in a 48-well plate, in which each PCR well contained 7 μl DNase/RNase free water, 1 μl of each primer (Table 1) (i.e., 1 μl of forwarding primer, and 1 μl of reverse primer), 1 μl of cDNA, and 10 μl of SYBR Green (Sigma-Aldrich, Germany). Thermal cycling was performed as described on the Applied Biosystems [25] sequence detection system. Pfafl method ($2^{-\Delta\Delta Ct}$, $\Delta\Delta Ct = \Delta C_{t\ Sample} - \Delta C_{t\ Control}$) was used to determine the fold change of each gene expression [26]. The primers used to assess the expression of Bax, Bcl-2, and GAPDH (as an internal control) genes possessed the following sequences [18]:

## Statistical analysis

SPSS software (version 21.0, Chicago, IL, USA) was used for data analysis. One-way analysis of variance (ANOVA) followed by a multiple two-tailed T-test was used to study significant differences among the collected data. $P < 0.05$ was considered statistically significant.

# 3. Results

## Biochemical profiles of samples

Fasting blood glucose levels significantly increased in the diabetic group compared to the control group (P<0.05). The comparisons showed a significant decrease in the intervention group compared to the control group (P<0.001) (Table 2). The serum insulin level significantly decreased in all the diabetic groups compared to the healthy groups (P<0.05). Meanwhile, serum insulin level increased in the intervention groups compared to the control groups (P<0.05) (Fig 1).

## Serum level of cholesterol

The serum cholesterol level significantly increased in the diabetic control group compared to the healthy groups (P<0.05). Moreover, the cholesterol level significantly decreased in all the groups treated with crocin and treadmill exercise compared to the diabetic control group (P<0.05) (Table 3). The serum level of LDL-C showed a slight increase in the diabetic control group compared to the healthy groups (P>0.05). The level of LDL-C was significantly reduced in the diabetic groups treated with crocin and exercise compared to the diabetic control group (P<0.05) (Table 3). The serum level of HDL-C significantly decreased in the diabetic control group compared to the healthy group (P<0.05). Also, it was significantly increased in the diabetic-treated groups compared to the diabetic control group (P<0.05). The level of HDL-C showed an insignificant increase in the diabetic-treated groups compared to the control group (P>0.05) (Table 3).

## Serum level of triglycerides (TG)

The serum level of TG significantly increased in the diabetic rats compared to the healthy group (P<0.05). Moreover, the level of TG significantly decreased in the treated rats compared to the diabetic control group (P<0.05) (Table 2). The serum level of creatinine significantly increased in the diabetic groups compared to the healthy groups (P<0.05). Also, the level of creatinine significantly decreased in the diabetic treated rats compared to the diabetic control group (P<0.05) (Fig 2). The serum level of urea significantly increased in the diabetic groups compared to the healthy groups (P<0.05). Furthermore, the level of urea significantly

**Table 2. The comparison of fasting blood glucose level in the study groups.** The asterisk (*) symbol shows significant difference between the control and diabetic groups and the plus (+) symbol shows significant difference between diabetic group and diabetic treatment groups ($P<0.05$). (C: Crocin, E: Exercise).

| Groups | One week before | One week after diabetes | At the end of study |
|---|---|---|---|
| | Mean ± SD | Mean ± SD | Mean ± SD |
| Control | 95.5± 3.34 | 98.32± 2.25+ | 95.4±3.34+ |
| Diabetic | 97.2± 2.6 | 305.25± 4.57* | 367.6±38.73* |
| Diabetic + Crocin | 98.5 ± 5.89 | 290.5± 4.93†* | 156.2±6.7+ |
| Diabetic + Exercise | 96.5 ± 4.5 | 295.82± 3.70+* | 134.2±15.38+ |
| Diabetic + C + E | 97.65± 2.6 | 302.5± 5.2+ | 122.75±6.1+ |
| Crocin | 95.60± 7.6 | 94.25± 3.7+ | 89.8±3.21+ |
| Exercise | 94.25± 2.6 | 93.37± 7.2+ | 90.8±4.21+ |
| Crocin + Exercise | 96.45 ± 3.4 | 95.8± 6.7+ | 92.3±8.05+ |

decreased in the diabetic rats treated with crocin and exercise compared to the diabetic control group (P<0.05) (Fig 3).

## Histopathological change in heart tissue

Based on the results of this study, in the control diabetic group, a small number of cardiomyocyte fibers were torn, and lymphocytes were observed in some fibers, which was less than in the control groups (Fig 4). The muscle fibers were connected irregularly with an indeterminate structure through the interlocking discs. The diameter of muscle fibers in this group was smaller than the healthy control group. Other studies also showed irregularities and accumulation in the nuclei and staining of the nuclei, which indicates the onset of apoptosis. The crocin-treated and regular exercise (group H) significantly reduced these changes and injuries.

## Changes in the antioxidant status

To examine the efficacy of crocin and treadmill exercise on oxidative stress induced by diabetes in the heart tissue, oxidative stress parameters, including MDA, SOD, and GPx, were

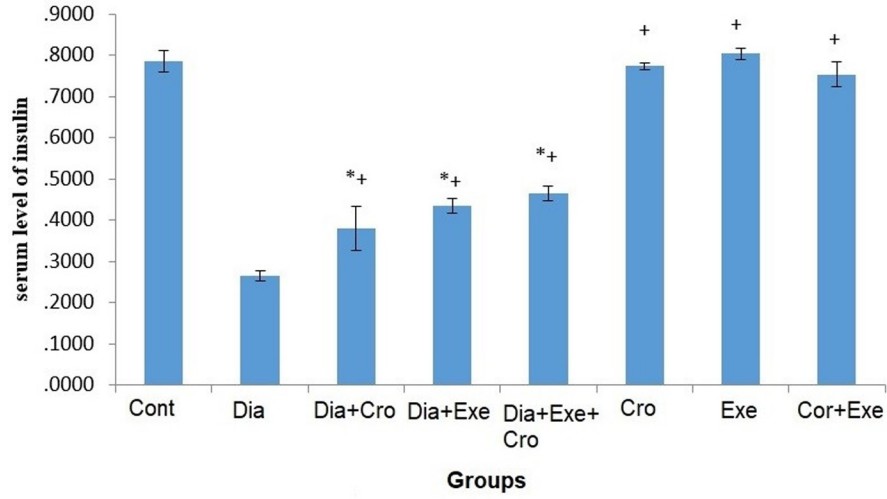

**Fig 1. The comparison of serum levels of insulin in the study groups.** The asterisk (*) symbol shows significant difference with the control group and the plus (+) symbol indicates a significant difference with the diabetic group ($P \leq 0.05$). (Cont: Control, Dia: Diabetic, Cro: Crocin, Exe: Exercise). Error bar type: SD: Standard Deviation.

**Table 3. The comparison of serum levels of lipid profile in the study groups.** The asterisk (*) symbol shows significant difference between control and diabetic groups. The plus (+) symbol shows significant difference between diabetic and diabetic treatment groups (P<0.05). (All measurements were done after the exercise and use of crocin.) (C: Crocin, E: Exercise).

| Groups | Cholesterol (Mg/dl) | HDL (Mg/dl) | LDL (Mg/dl) | TG (Mg/dl) |
| --- | --- | --- | --- | --- |
| | Mean ± SD | Mean ± SD | Mean ± SD | Mean ± SD |
| Control | 56. 50±2.32 | 45.68±1.59 | 18.74±1.51 | 34.55±1.58 |
| Diabetic | 80.89±1.04+ | 20.92±0.53+ | 32.78±1.68 | 70.37±2.58+ |
| Diabetic + Crocin | 72. 26±2.30* | 35.76±2.61* | 26.91±1.99* | 48.03±1.59* |
| Diabetic + Exercise | 65.56±1.45* | 36.24±1.20* | 24.52±1.15* | 45.95±1.23* |
| Diabetic + C + E | 62.52±1.37* | 37.13±1.18* | 22.12±1.03* | 44.75±2.53* |
| Crocin | 54.15±2.17* | 43.24±1.12* | 15.42±1.05* | 35.25±1.73* |
| Exercise | 55.12±2.27* | 45.15±1.15* | 16.22±1.13* | 34.78±2.13* |
| Crocin + Exercise | 52.22±2.17* | 44.23±1.25* | 15.62±1.03* | 32.85±2.23* |

assessed (Tables 4 and 5). As can be seen, diabetes in the diabetic control group led to a notable reduction in antioxidant enzyme level, SOD, and GPx compared to the control group ($P$ <0.05), while treatment with crocin and treadmill exercise remarkably enhanced the activity of these enzymes compared to the diabetic control group ($P$ <0.05). Although diabetes significantly improved the concentration of MDA in the heart tissues compared to the control group ($P$ ≤0.03), crocin treatment and treadmill exercise resulted in a decline in lipid peroxidation levels compared to the diabetic control group ($P$ <0.05). However, comparing all oxidative stress parameters between the control and treated-healthy groups revealed no significant changes ($P$ >0.05).

## Relative expression of Bax and Bcl-2

RT-qPCR was exerted to examine the effect of crocin and treadmill exercise on the mRNA expression of Bax and Bcl-2 genes in the heart tissue of diabetic rats. Diabetes significantly enhanced the mRNA expression of Bax compared to the control group ($P$ <0.001) (Table 6). In diabetic animals, the ratio of Bcl-2 mRNA was significantly reduced compared to the

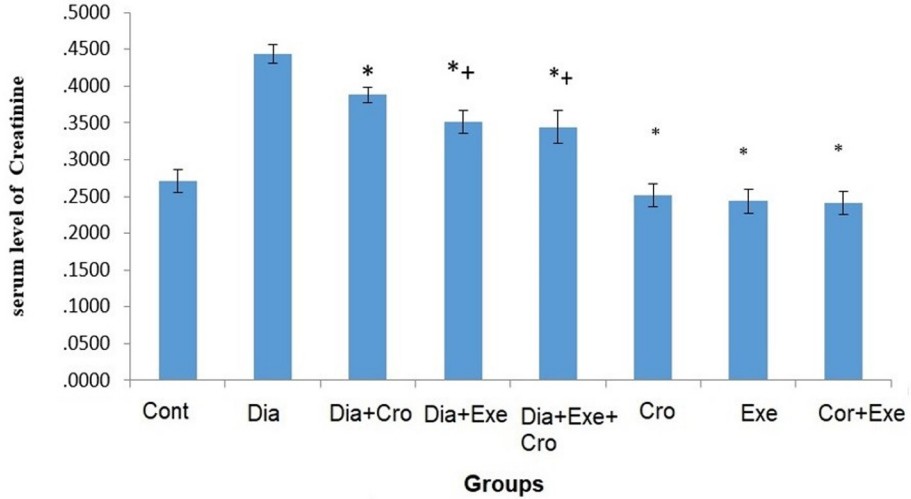

**Fig 2. The comparison difference with the diabetic control group.** The plus (+) symbol indicates significant difference with the control group ($P$≤0.05) of serum levels of Creatinine in the study groups. The asterisk (*) symbol shows significant. (Cont: Control, Dia: Diabetic, Cro: Crocin, Exe: Exercise). Error bar type: SD: Standard Deviation.

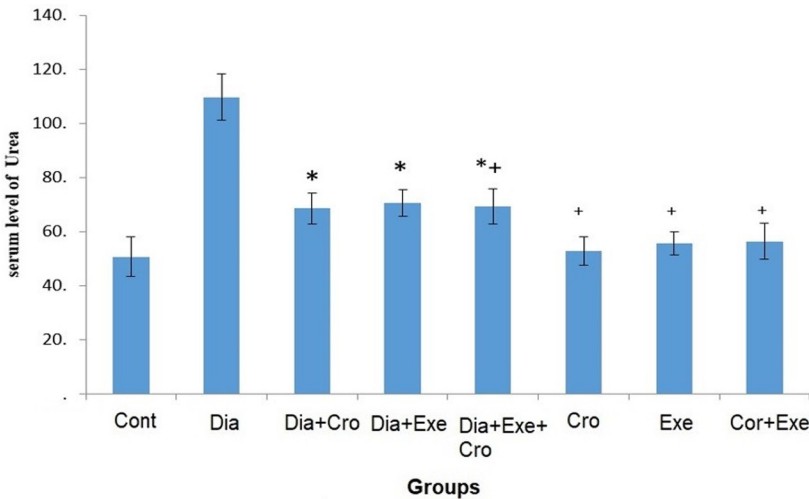

**Fig 3. The comparison of serum levels of urea in the study groups.** The asterisk (*) symbol shows significant difference with the control group. The plus (+) symbol indicates significant difference with the diabetic control group (*P*≤0.05). (Cont: Control, Dia: Diabetic, Cro: Crocin, Exe: Exercise). Error bar type: SD: Standard Deviation.

control group ($P < 0.002$), but treatment with crocin and treadmill exercise and a combination of them raised the mRNA ratio of Bcl-2 compared to the diabetic control group ($P < 0.05$). There was no significant difference between the healthy control and treated-healthy control groups regarding both genes ($P > 0.05$). Furthermore, the Bax/Bcl-2 ratio was significantly increased in the diabetic control group compared to the control group ($P \leq 0.001$), while it was significantly decreased in the treatment groups ($P \leq 0.001$). However, there was no significant difference between the healthy control group and the treated-healthy control group ($P > 0.05$).

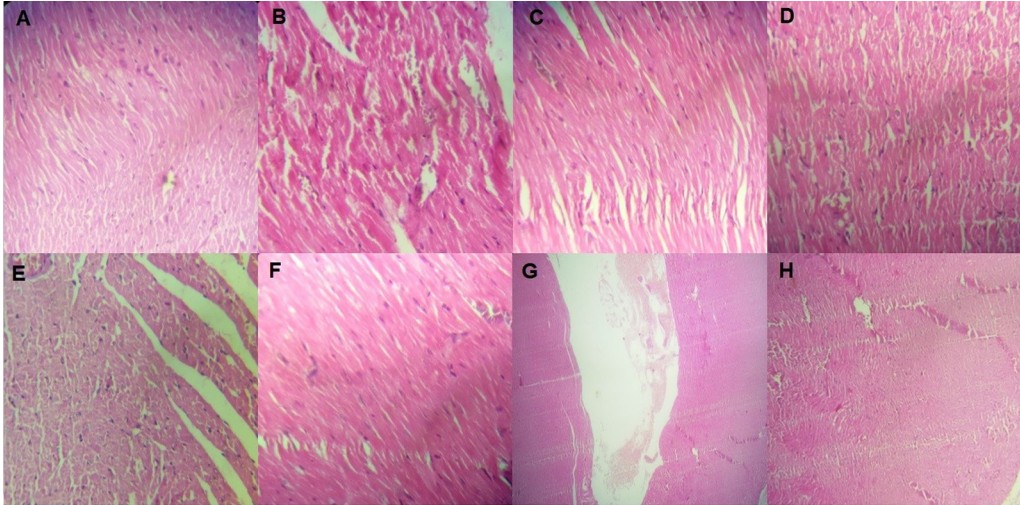

**Fig 4. Histological change in study groups.** A. Control; B. Diabetic; C. Diabetic + crocin; D. Diabetic + exercise; E. Diabetic + Crocin + Exercise; F. Crocin; G. Exercise; and H. Crocin + Exercise. As shown in the figure, crocin-treated and regular exercise (group H) significantly reduce the changes and injuries.

**Table 4. The concentration of serum oxidative stress markers in the study groups.** Data are shown as Mean ± (SE). * in Comparison with control group (P = 0.001) and + in Comparison with diabetic group (P = 0.001). (All measurements were done after the exercise and use of crocin.) (C: Crocin, E: Exercise).

| Groups | MDA Mean (±) SE | GPX Mean (±) SE | SOD Mean (±) SE |
|---|---|---|---|
| Control | 0.62 ± 0.09 | 5.85 ± 0.057 | 1.63 ± 0.21 |
| Diabetic | 1. 85 ± 0.25* | 2.88 ± 0.048 * | 0.60± 0.11* |
| Diabetic + Crocin | 0.97 ± 0.15*+ | 4.22± 0.072 + | 1.16± 0.18*+ |
| Diabetic + Exercise | 1.00 ± 0.11* | 3.95± 0.10 + | 1.37± 0.24*+ |
| Diabetic + C + E | 0.92 ± 0.09* | 4.32± 0.084 + | 1.27± 0.14*+ |
| Crocin | 0.70 ± 0.08+ | 5.97 ± 0.037+ | 1.63± 0.21+ |
| Exercise | 0.72 ± 0.12+ | 5.50 ± 0.077+ | 1.60± 0.15+ |
| Crocin + Exercise | 0.68 ± 0.15+ | 6.05 ± 0.059 + | 1.65± 0.25+ |

## 4. Discussion

DM is a metabolic disorder characterized by hyperglycemia and insufficient insulin secretion or function. Even though the primary etiology of this disease is unknown, the available information indicates the role of viral infection, autoimmune disease, and environmental factors in its development [2, 3]. Despite controlling several complications of diabetes by exogenous insulin, multiple complications of this disease are common in the cardiovascular system, kidney, retina, lens of the eye, peripheral nerves, and skin. So, preventing and controlling these complications is essential for having a long life with high quality. Diabetes is a serious cause of cardiovascular disease in humans. It can also cause disability and increase mortality in patients with cardiovascular disease. Following the onset of the disease, hyperglycemia is the leading cause of most chronic complications, including cardiac tissue in patients with interstitial metabolism disorders, fibrosis, impaired cellular function in vascular smooth muscle, and impaired contractile function [6]. Increased oxidative stress has been widely accepted as one factor in the development and progression of diabetes and its complications. Diabetes is usually associated with increased free radicals or impaired antioxidant defense production. DM and its complications in various body organs, especially in the heart, are related to the accumulation of free radicals. The increased oxidative stress leads to the activation of transcription factors, increased glucose oxidation [27], and increased synthesis of end products resulting from advanced glycation [8]. Intense physical exercise has been shown to increase the production of free radicals, resulting in cellular damage to body tissues. Meanwhile, regular and moderate-intensity physical activity can strengthen the antioxidant system against diseases [28].

**Table 5. The concentration of oxidative stress markers in heart tissue.** Data are shown as Mean ± (SE). * in comparison with control group (P = 0.001) and + in comparison with diabetic group (P = 0.001). (All measurements were done after the exercise and use of crocin.) (C: Crocin, E: Exercise).

| Groups | MDA ± SD (nmol/g wet tissue) | SOD ± SD (Iv/gr wet tissue) | GPx ± SD ($I_{V}$/gr wet tissue) |
|---|---|---|---|
| Control | 78.25 ± 6.5 | 1.63± 0.21 | 33 ± 3.21 |
| Diabetic | 148.65 ± 7.35* | 0.60± 0.11* | 14.35 ± 2.3* |
| Diabetic + Crocin | 97.75 ± 5.7*+ | 1.16± 0.18*+ | 24.75 ± 3.6*+ |
| Diabetic + Exercise | 100 ± 5.25* | 1.37± 0.24*+ | 21.65 ± 3.25*+ |
| Diabetic + C + E | 95 ± 2.25* | 1.27± 0.14*+ | 19.80 ± 3.25*+ |
| Crocin | 70.35 ± 6.5 | 1.63± 0.21 | 33 ± 3.21 |
| Exercise | 75.25 ± 2.2+ | 1.60± 0.15+ | 35.56 ± 4.2+ |
| Crocin + Exercise | 80.15 ± 3.5+ | 1.65± 0.25+ | 30.65 ± 2.25+ |

**Table 6. The Bax and BCL-2 gene expressions.** Data are shown as Mean ± (SE). *in comparison with control group ($P = 0.001$) and + in comparison with diabetic group ($P = 0.001$). (All measurements were done after the exercise and use of crocin.) (C: Crocin, E: Exercise).

| Group | Bax | Bcl-2 | Bax/Bcl-2 ratio |
|---|---|---|---|
| Groups | 0.42± .022 | 1.38± 0.034† | 0.30±0.31† |
| Control | 1.38± .032* | 0.41± 0.038* | 4.56±0.051* |
| Diabetic | 1.01±.04 | 0.64± 0.014† | 1.57±4.25† |
| Diabetic + Crocin | 1.05±.0.32† | 0.62± 0.011† | 1.69±0.025† |
| Diabetic + Exercise | 0.39± .068 | 1.51± 0.073† | 0.25±0.062† |
| Diabetic + C + E | 0.44±.012† | 1.40± 0.028† | 0.31±0.045† |
| Crocin | 0.45±.016† | 1.35± 0.022† | 0.33±0.051† |
| Exercise | 0.45±.014† | 1.42± 0.028† | 0.35±0.025† |

Currently, the existing drug regimens to control diabetes have some drawbacks. So, safer and more efficient anti-diabetic drugs should be developed [29]. Diabetes causes a disorder in glucose absorption and its metabolism. A single dose of streptozotocin (STZ) as low as 50 mg/kg produces incomplete destruction of pancreatic beta cells, even though the rats become permanently diabetic [30]. This study aimed to evaluate the effectiveness of crocin and treadmill exercise on biochemical blood factors associated with diabetes in diabetic rats induced by STZ. The results showed that intraperitoneal injections of 50 mg/kg STZ caused hyperglycemia and diabetes in rats within 72 hours. Chopra et al. showed that 50 mg/kg STZ could induce hyperglycemia and diabetes within 72 hours [31]. Also, Petkov et al. used an intraperitoneal injection of STZ dissolved in citrate buffer (pH = 4.5) to induce diabetes within 72 hours [32].

In the present study, after confirming hyperglycemia in rats, we used crocin and exercise on a treadmill to control blood sugar; our treatment method significantly influenced the blood sugar in diabetic rats. The anti-hyperglycemic effects of crocin include antioxidant and anti-inflammatory properties, flavonoids, and phenols [33]. According to our results, the serum glucose level significantly decreased in treadmill exercise groups. In this context, G. Reuter *et al*. showed that exercise on the treadmill reduced blood surge [34]. The comparison of serum insulin levels in different groups showed that STZ-induced diabetes caused damage to pancreatic beta cells and reduced blood insulin levels [32, 35]. In a study, P. Palit *et al*. showed that STZ damaged the beta cells of the pancreas and decreased the serum levels of insulin [36]. Treatment with crocin and treadmill exercise can increase insulin levels, which may be due to the presence of compounds such as flavonoids and stimulation of the pancreas by training [37, 38]. Therefore, it has a high antioxidant property and can protect beta cells against oxidative stress [39]. Shu et al. showed that flavonoids could lower blood sugar, protect the beta cell against oxidative stress, and maintain pancreatic beta cell integrity [40].

Moreover, other studies suggested that antioxidants protect the beta cells against oxidative stress and prevent cell destruction [7, 41]. Diabetes is a disease associated with metabolism, and it can cause a disturbance in lipid metabolism [27]. In the present study, by comparing the lipid profiles in different groups, we found that diabetes can increase cholesterol, LDL-C, and triglyceride levels, while decreasing HDL-C levels. This is in line with Rasekh *et al*. [42]. In our study, the level of LDL-C in the diabetic control group insignificantly increased compared to the control group. The serum level of the lipid profile could be controlled during treatment with crocin and treadmill exercise [43]. These plant extracts significantly reduced TG, cholesterol, and LDL-C levels while increasing HDL-C. This might be due to having abundant antioxidant compounds such as flavonoids [42], which can control blood lipid levels [44].

Studies in animal models showed the positive effects of regular and moderate-intensity exercise during experimental diabetes. Naderi et al. showed that voluntary exercise diminishes the MDA level in the blood and heart tissue of diabetic rats. It also accentuates activities of SOD, GPX, and CAT. Therefore, it may be considered a valuable tool for reducing oxidative stress in diabetes [45]. Farshid et al. showed cardioprotective effects of crocin and insulin in STZ-induced diabetic rats. The antioxidant and anti-hyperglycemic properties of crocin and insulin may be involved in their cardioprotective actions [46]. Regular and medium-intensity physical exercise can increase the activity of antioxidant enzymes involved in defending against stressors and protecting the body's organs from complications of diabetes [47]. Due to the controversial results of studies on the effects of diabetes on antioxidant status, this study aimed to evaluate this issue. Another study by Kayama *et al.* found that oxidative stress in heart tissue caused by diabetes led to tissue damage, degradation, and reduced function [48]. Activated species of foreign oxygen generated by binding of TNF-α and FasL to their known receptors (TNFR1 and Fas, respectively) can activate the external apoptotic pathway induced by the initiator caspase-8, which in turn activates downstream pathways, leading to activation of caspase-3 or -7, which are executive caspases, resulting in apoptosis [49]. In addition, apoptosis in cardiac cells can stimulate the release of cytochrome C from the mitochondrial inner membrane to the cytoplasm through the endogenous apoptosis pathway, resulting in the accumulation of danger signals, such as the overproduction of ROS. It leads to the formation of a complex protein called the apoptosome, which can activate procaspase 9 [50]. Yaribeygi et al. showed that crocin can prevent and improve DN through three distinct mechanisms. First, crocin potentiates the antioxidative defense system and scavenges free radicals, attenuating oxidative stress. Second, crocin inhibits inflammatory reactions by lowering IL-18 cytokine expression, an upstream inflammatory agent that plays a vital role in the onset of the inflammatory cascade in DN; so, crocin can prevent inflammation-induced DN damage. Third, apoptotic processes, which are crucial for DN-induced tissue injuries, are weakened by crocin. Therefore, we conclude that crocin can prevent structural and cellular damage observed in uncontrolled diabetes in kidneys and improve renal function [51].

The internal pathway of apoptosis is precisely regulated by two main groups of proteins responsible for apoptosis activation/inhibition. Some molecules, including nitric oxide, Bax, Bak, and Bid, can activate pro-apoptotic pathways such as cytochrome C release, while Bcl-2 and Bcl-xL proteins inhibit apoptosis [52].

Several studies have reported that oxidative stress induced by ROS can increase the phosphorylation of the Bcl-2 family, thus shifting the balance of pro-apoptotic and anti-apoptotic agents in favor of synthesizing apoptotic proteins such as Bax. We showed that Bax gene expression was significantly higher in the diabetic group than in the control group. Conversely, Bcl-2 expression was significantly lower in the genes of the diabetic group than in the control group. Studies have shown that diabetes increases oxidative stress, induces apoptosis in heart tissue, and increases Bax and decreases Bcl-2 expression [53]. Reduced blood flow increases nitric oxide production, resulting in tissue damage. Several studies indicated the protective effects of regular exercise against cardiovascular disease following increased age and reduced mortality due to disorders of this organ [54]. Cellular mechanisms involved in applying these protective effects have not been fully identified, and the results of studies are different in this regard. However, regular and moderate-intensity exercise is used as a preventive tool by a large number of health professionals [55].

It should be noted that the administration of crocin and continuous exercise did not significantly affect cardiac tissue, indicating that crocin and constant exercise are a safe combination. However, our findings were consistent with the results of a study by Demir *et al.*, reporting that crocin could reduce the degeneration of myocardial cells and damage caused by

myocardial infarction [56]. Previous studies have shown that using antioxidants can reduce the destructive effects of oxidative stress on heart tissue. In this regard, AL. hashem *et al*. showed that using vitamins E and C can reduce the impact of oxidative stress on the cardiovascular system [57]. Another study by Karajibani et al. found that using grape seed oil as an antioxidant could reduce the oxidative stress induced by myocardial infarction and prevent damage to heart tissue. The study also found that using antioxidants in grape seed oil prevented lipid peroxidation, a factor in damage induction. Also, this antioxidant increased the activity of antioxidant enzymes such as SOD and GPx. This indicates that using antioxidants strengthens the antioxidant defense system [58].

Therefore, crocin administration and regular exercise significantly reduced lipid peroxidation (MDA) levels. Using crocin and regular exercise also increased the levels of antioxidant enzymes such as SOD and GPx, thereby strengthening the antioxidant defense system. Studies have shown that antioxidant enzymes can neutralize the harmful effects of free radical molecules by suppressing those molecules and protecting heart tissue against oxidative stress [59].

Studies have shown crocin can inhibit the progression of apoptosis by an increase in antioxidant enzyme levels. This could inhibit the release of cytochrome C, Apaf 1, and pro-caspase 9 in the cells because the release of cytochrome C can increase pro-caspase-9. All of this can increase produce subtypes of caspase-3 as an effective factor in increasing the Bax and P53 and decreasing Bcl-2 levels [60]. Different studies have been carried out on the effects of crocin and exercise on apoptosis in animal cells. For example, crocin can increase the death of tumor cells in lung and breast cancers [60, 61]. Another study showed crocin can significantly reduce P53 levels in the heart tissue of diabetic rats [62]. Our findings showed that the Bax/Bcl-2 ratio was significantly higher in the control diabetes group than in the sham group. At the same time, crocin treatment and regular exercise moderated this increase relative to baseline. Previous studies have shown that crocin and regular exercise can reduce the expression of pro-apoptotic agents and increase the expression of anti-apoptotic proteins [63]. Our research found that the Bax/Bcl2 ratio in the crocin-treated and regular exercise groups was significantly lower than in the control diabetes group.

On the other hand, in contrast to Bax expression, studying the expression of the Bcl-2 gene in control diabetic group showed that the expression of this gene was significantly reduced compared to the sham group. However, the Bcl-2 mRNA and protein levels in the diabetic groups treated with crocin and regular exercise increased compared to the diabetes group. Recent studies have suggested that the Bax/Bcl-2 ratio can determine the fate of cells and whether or not they undergo apoptosis [64]. Due to the value of the Bax/Bcl-2 ratio, the apoptosis rate due to oxidative stress in the heart tissue of diabetic rats was evaluated. Evidence suggests that Bcl-2 can act as a potent antioxidant. Studies have shown that Bcl-2 controls antioxidant defense through lipid peroxidation or without changes in the contents of the active species of intracellular oxygen. The anti-apoptotic effects of crocin as an antioxidant molecule and increased Bcl-2 protein expression have been demonstrated in different studies [65, 66].

## 5. Conclusions

The present study showed that diabetes increased oxidative stress and cardiac tissue damage in diabetic rats so that the MDA level in the blood serum of diabetic rats and their heart tissue significantly increased. Also, the levels of anti-enzyme oxidation of SOD and GPx decreased in diabetic groups. Moreover, by examining the heart tissue, we found that tissue damage in the groups with diabetes significantly increased, and the expression of the Bax gene in this group was substantially higher than in the control group. This study showed that co-administration

of crocin and regular exercise alone as an antioxidant reduced the destructive effects of diabetes on heart tissue structure in rats.

## Acknowledgments

The authors wish to thank the Deputy for Research and Technology of the Islamic Azad University of Tabriz for support of the study. We also thank the Clinical Research Unit of Alzahra Hospital in Tabriz. The ethics committee of the Islamic Azad University of Tabriz approved the study protocol (code: IR.IAU.TABRIZ.REC.1401.005).

## Author Contributions

**Data curation:** Laleh Pourmousavi, Rasoul Hashemkandi Asadi, Farzad Zehsaz.

**Funding acquisition:** Farzad Zehsaz.

**Investigation:** Laleh Pourmousavi, Rasoul Hashemkandi Asadi.

**Methodology:** Laleh Pourmousavi, Farzad Zehsaz.

**Software:** Roghayeh Pouzesh Jadidi.

**Supervision:** Roghayeh Pouzesh Jadidi.

**Validation:** Roghayeh Pouzesh Jadidi.

**Visualization:** Roghayeh Pouzesh Jadidi.

**Writing – original draft:** Rasoul Hashemkandi Asadi.

**Writing – review & editing:** Laleh Pourmousavi, Rasoul Hashemkandi Asadi.

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
