## [Editor Report · Decision Letter 0]

5 Feb 2023

PONE-D-23-00007Effect of crocin and treadmill exercise on oxidative stress and heart damage in diabetic ratsPLOS ONE

Dear Dr. hashemkandi,

Thank you for submitting your manuscript to PLOS ONE. After careful consideration, we feel that it has merit but does not fully meet PLOS ONE’s publication criteria as it currently stands. Therefore, we invite you to submit a revised version of the manuscript that addresses the points raised during the review process. Please submit your revised manuscript by Mar 22 2023 11:59PM. If you will need more time than this to complete your revisions, please reply to this message or contact the journal office at plosone@plos.org. Please include the following items when submitting your revised manuscript:A rebuttal letter that responds to each point raised by the academic editor and reviewer(s). You should upload this letter as a separate file labeled 'Response to Reviewers'.A marked-up copy of your manuscript that highlights changes made to the original version. You should upload this as a separate file labeled 'Revised Manuscript with Track Changes'.An unmarked version of your revised paper without tracked changes. You should upload this as a separate file labeled 'Manuscript'.

We look forward to receiving your revised manuscript.

Kind regards,

Wajdy Jum’ah Al-Awaida, Ph.D

Academic Editor

PLOS ONE

Journal Requirements:

a) The name of the colleague or the details of the professional service that edited your manuscript.

b) A copy of your manuscript showing your changes by either highlighting them or using track changes (uploaded as a *supporting information* file).

c) A clean copy of the edited manuscript (uploaded as the new *manuscript* file).

4. We noticed you have some minor occurrence of overlapping text with the following previous publication(s), which needs to be addressed:

- http://eprints.lums.ac.ir/1036/1/Protective%20Effect%20of%20Galega%20officinalis%20Extract%20on%20Streptozotocin-Induced%20Kidney%20Damage%20and%20Biochemical%20Factor%20in%20Diabetic%20Rats.pdf?

- https://www.sciencedirect.com/science/article/pii/S0753332218358712?via%3Dihub

In your revision ensure you cite all your sources (including your own works), and quote or rephrase any duplicated text outside the methods section. Further consideration is dependent on these concerns being addressed.

7. PLOS requires an ORCID iD for the corresponding author in Editorial Manager on papers submitted after December 6th, 2016. Please ensure that you have an ORCID iD and that it is validated in Editorial Manager. To do this, go to ‘Update my Information’ (in the upper left-hand corner of the main menu), and click on the Fetch/Validate link next to the ORCID field. This will take you to the ORCID site and allow you to create a new iD or authenticate a pre-existing iD in Editorial Manager. Please see the following video for instructions on linking an ORCID iD to your Editorial Manager account: https://www.youtube.com/watch?v=_xcclfuvtxQ

8. Please ensure that you refer to Figure 4 in your text as, if accepted, production will need this reference to link the reader to the figure.

9. We note you have included a table to which you do not refer in the text of your manuscript. Please ensure that you refer to Table 1, 5, and 6 in your text; if accepted, production will need this reference to link the reader to the Table.

**Additional Editor Comments:**

Dear Author,

I am writing to suggest corrections to an article that was recently published in your journal. The following comments should be corrected:

1. The total RNA of heart tissues was isolated, not genes.

2. Figure 4 needs more resolution and magnification power.

3. In Figure 1, Figure 2, and Figure 3, "crosin" should be replaced by "crocin".

4. The authors should provide more information about the aerobic exercise regimes and the crocin administration dosages.

5. The authors should also provide a discussion of the potential mechanisms by which aerobic exercise and crocin administration reduce tissue damage, oxidative stress, and gene expression associated with apoptosis control.

Sincerely,

---

## [Author Response · Author response to Decision Letter 0]

10 Jul 2023

PONE-D-23-00007

Effect of crocin and treadmill exercise on oxidative stress and heart damage in diabetic rats

PLOS ONE

Dear Dr. hashemkandi,

Thank you for submitting your manuscript to PLOS ONE. After careful consideration, we feel that it has merit but does not fully meet PLOS ONE’s publication criteria as it currently stands. Therefore, we invite you to submit a revised version of the manuscript that addresses the points raised during the review process.

We look forward to receiving your revised manuscript.

Kind regards,

We would like to thank the reviewers for their constructive comments and for the positive opinion of our manuscript. We have revised the manuscript according to the critiques raised by the reviewers. Highlighted texts in the revised manuscript indicate the changes we have made. The following are the point-by-point responses to the reviewers. 

Wajdy Jum’ah Al-Awaida, Ph.D

 Academic Editor

PLOS ONE

Journal Requirements:

Thanks for your attention. The paper was formatted.

Thanks for your good comment. The required part was added to the manuscript and highlighted.

To induce type 1 diabetes; streptozotocin (STZ) was injected via the tail vein at a dose of 50 mg/kg in 0.1 mol/L citrate buffer (pH 4.5) or citrate buffer alone as control under anesthesia . One week after the STZ injection, rats exhibiting hyperglycemia were considered type 1 diabetic and subjected to subsequent experiments. After the completion of experiments, rats were deeply anesthetized with sodium pentobarbital (60 mg/kg), and hearts were rapidly excised and frozen in liquid nitrogen for later analysis.

Thanks for your relevant and important comment. The language and grammar of the paper were checked. The certificate of that was attached. 

a) The name of the colleague or the details of the professional service that edited your manuscript.

b) A copy of your manuscript showing your changes by either highlighting them or using track changes (uploaded as a *supporting information* file).c) A clean copy of the edited manuscript (uploaded as the new *manuscript* file).

4. We noticed you have some minor occurrence of overlapping text with the following previous publication(s), which needs to be addressed:

- http://eprints.lums.ac.ir/1036/1/Protective%20Effect%20of%20Galega%20officinalis%20Extract%20on%20Streptozotocin-Induced%20Kidney%20Damage%20and%20Biochemical%20Factor%20in%20Diabetic%20Rats.pdf?

- https://www.sciencedirect.com/science/article/pii/S0753332218358712?via%3Dihub

In your revision ensure you cite all your sources (including your own works), and quote or rephrase any duplicated text outside the methods section. Further consideration is dependent on these concerns being addressed.

Thanks for your relevant and important comment. The Plagiarism in the paper was double-checked and revised in similar parts.

Thanks for your relevant and important comment. Supporting data (include raw data) the findings of current study are available from corresponding author on request. 

Thanks for your relevant and important comment. Supporting data (include raw data) the findings of current study are available from corresponding author on request. 

7. PLOS requires an ORCID iD for the corresponding author in Editorial Manager on papers submitted after December 6th, 2016. Please ensure that you have an ORCID iD and that it is validated in Editorial Manager. To do this, go to ‘Update my Information’ (in the upper left-hand corner of the main menu), and click on the Fetch/Validate link next to the ORCID field. This will take you to the ORCID site and allow you to create a new iD or authenticate a pre-existing iD in Editorial Manager. Please see the following video for instructions on linking an ORCID iD to your Editorial Manager 

account: https://www.youtube.com/watch?v=_xcclfuvtxQ

8. Please ensure that you refer to Figure 4 in your text as, if accepted, production will need this reference to link the reader to the figure.

Thanks for your comments. It is linked to the text. 

9. We note you have included a table to which you do not refer in the text of your manuscript. Please ensure that you refer to Table 1, 5, and 6 in your text; if accepted, production will need this reference to link the reader to the Table.

Thanks for your comments. Those were linked to the text. 

Additional Editor Comments:

Dear Author,

I am writing to suggest corrections to an article that was recently published in your journal. The following comments should be corrected:

1. The total RNA of heart tissues was isolated, not genes.

Thanks for your attention. It was revised.

2. Figure 4 needs more resolution and magnification power.

Thanks for your attention. It was replaced.

3. In Figure 1, Figure 2, and Figure 3, "crosin" should be replaced by "crocin". 

Thanks for your attention. Those were corrected.

4. The authors should provide more information about the aerobic exercise regimes and the crocin administration dosages. 

Exercise and crocin administration protocol: In the first stage (first to fourth weeks), the rats ran on the treadmill for 60 minutes with a speed of 11.75-7.83 m/min (equivalent to 40-60% VO2 max), and in the second stage (fifth to eighth weeks) they ran with a speed of 10/55-83 m/min (equivalent to 40-60% of VO2 max) five sessions per week. To achieve adaptation to the exercises, we increased the intensity and duration of the training. Each training session consisted of three steps: 1) warm-up (5 minutes with 30% VO2 max), 2) MICT (for 60 minutes with 40-60% VO2 max), and 3) return to the original state (5 minutes with 30% VO2 max). And also crocin administration dosages was 50 mg/kg.

5. The authors should also provide a discussion of the potential mechanisms by which.

Studies have shown crocin can inhibit the progression of apoptosis by an increase in antioxidant enzyme levels. This could inhibit the release of cytochrome C, Apaf 1, and pro-caspase 9 in the cells because the release of cytochrome C can increase pro-caspase-9. All of this can increase produce subtypes of caspase-3 as an effective factor in increasing the Bax and P53 and decreasing Bcl-2 levels. Different studies have been carried out on the effects of crocin and exercise consumption on apoptosis in animal cells. For example, crocin can increase the death of tumor cells in lung and breast cancers. Another study showed crocin can significantly reduce P53 levels in the heart tissue of diabetic rats.

Sincerely,

---

## [Decision Letter · Decision Letter 1]

10 Aug 2023

PONE-D-23-00007R1Effect of crocin and treadmill exercise on oxidative stress and heart damage in diabetic ratsPLOS ONE

Dear Dr. hashemkandi,

Thank you for submitting your manuscript to PLOS ONE. After careful consideration, we feel that it has merit but does not fully meet PLOS ONE’s publication criteria as it currently stands. Therefore, we invite you to submit a revised version of the manuscript that addresses the points raised during the review process.Please submit your revised manuscript by Sep 24 2023 11:59PM. If you will need more time than this to complete your revisions, please reply to this message or contact the journal office at plosone@plos.org. Please include the following items when submitting your revised manuscript:A rebuttal letter that responds to each point raised by the academic editor and reviewer(s). You should upload this letter as a separate file labeled 'Response to Reviewers'.A marked-up copy of your manuscript that highlights changes made to the original version. You should upload this as a separate file labeled 'Revised Manuscript with Track Changes'.An unmarked version of your revised paper without tracked changes. You should upload this as a separate file labeled 'Manuscript'.If applicable, we recommend that you deposit your laboratory protocols in protocols.io to enhance the reproducibility of your results. Protocols.io assigns your protocol its own identifier (DOI) so that it can be cited independently in the future. For instructions see: https://journals.plos.org/plosone/s/submission-guidelines#loc-laboratory-protocols. Additionally, PLOS ONE offers an option for publishing peer-reviewed Lab Protocol articles, which describe protocols hosted on protocols.io. Read more information on sharing protocols at https://plos.org/protocols?utm_medium=editorial-email&utm_source=authorletters&utm_campaign=protocols.

We look forward to receiving your revised manuscript.

Kind regards,

Wajdy Jum’ah Al-Awaida, Ph.D

Academic Editor

PLOS ONE

Journal Requirements:

Reviewers' comments:

Reviewer's Responses to Questions

**Comments to the Author**

1. If the authors have adequately addressed your comments raised in a previous round of review and you feel that this manuscript is now acceptable for publication, you may indicate that here to bypass the “Comments to the Author” section, enter your conflict of interest statement in the “Confidential to Editor” section, and submit your "Accept" recommendation.

Reviewer #1: (No Response)

Reviewer #2: (No Response)

2. Is the manuscript technically sound, and do the data support the conclusions?

Reviewer #1: Yes

Reviewer #2: Yes

3. Has the statistical analysis been performed appropriately and rigorously? 

Reviewer #1: Yes

Reviewer #2: Yes

4. Have the authors made all data underlying the findings in their manuscript fully available?

Reviewer #1: No

Reviewer #2: No

5. Is the manuscript presented in an intelligible fashion and written in standard English?

Reviewer #1: No

Reviewer #2: Yes

6. Review Comments to the Author

Reviewer #1: Dear Dr. Hashemkandi,

I appreciate the paper and results you have presented. Although they are not entirely original,

most of these insights have been described in previous publications by other research groups.

However, the data provided here support and complement the previous findings. I particularly

commend the simplicity and focus of the project, which was conducted entirely in vivo. The

study contributes to our understanding of cardiovascular defects associated with diabetes and

their contribution to the lethality in DM. It also explores new possibilities for patients to

mitigate the impact of reactive oxygen species (ROS) on cardiac tissue, thereby increasing the

potential to reduce risks. The authors should include references of important previous works in

these areas. For instance, a study from the same center demonstrated beneficial effects of

exercise on cardiovascular damage in diabetic rats (Naderi et al, 2015 Adv Pharm Bull).

Additionally, previous studies have shown crocin, an antioxidant agent presents in saffron,

reduces the impact of ROS in cardiovascular tissue in diabetics (Farshid et al, 2016, Avicenna J

Phytomed). The findings of these papers should be discussed on the manuscript.

Furthermore, it is essential to improve the English usage, grammar and vocabulary especially in

the abstract. I agree that revised version is improved respect to the first one.

Another general concern I have is regarding how the rat groups are described along the text.

To enhance readability, I suggest using “diabetes group”, “diabetes treated group

(exercise/crocin)” and “control/healthy group”, instead of “diabetes control group”.

I also concur previous reviewer’s suggestion to provide access to the raw data. I highly

recommend that the authors make it publicly available.

Here some additional points that I believe should be addressed:

1. I would like to know the exact number of rats in each of the 8 groups detailed in

“Animal preparation and study design” section of Material and Methods.

2. In table 3, 4, 5 and 6 please indicate when the samples for the measurements were

collected.

3. For figures 1, 2 and 3, as well as tables 2 and 3, clarify the representation of

numbers/bars (average, median, …) and the type of error bars used (SEM, SE,..).

4. Include reference to previous studies that have been assessed the effects of crocin and

exercise on generation of cardiovascular damage in diabetes models.

5. Consider improving the visual presentation of tables by using graphs, particularly for

tables 5 and 6.

6. In the histological studies, I highly recommend implementing a method to quantify the

extent of histological damage. This way you can strengthen the analysis and provide

more robust assessment of the observed histopathological changes. This also will

enhance the objectivity and reliability of the findings and will complement the

microscope images.

7. It would be beneficial to include comments in the text regarding the recovery of

kidney function after exercise and crocin treatment, supported by the levels of

creatinine and urea.

Additionally, I have some suggestions to enhance the comprehensiveness of the paper and

introduce further novelty by incorporating complementary readouts to the experiments:

1. In addition to measuring of MDA, SOD and GPx to asses oxidative stress, that is an

indirect way, consider including another assay, there are a bunch of kits that help to

confirm an increase of the oxidative stress, I will recommend to use one to check the

oxidation levels of the heart proteins in the 8 groups of rats.

2. To complement the histological approach and confirm apoptosis in a direct way,

consider conducting a functional assay, a very easy one will be to measuring annexin V

levels by flow cytometry in rat heart cells. Additionally, to investigate apoptosis

induction, which is determined by the balance of pro- and anti- apoptotic regulators,

consider measuring the levels of other apoptotic regulators by qPCR (Bcl-x, MCL1, BAD,

BIM,…) this way results would be more robust. Alternatively, BH3 profiling offers a

novel and interesting approach to test apoptosis.

In conclusion, I find the work very promising, and with the necessary improvements, I

would recommend that the editor accept it for publication in PLOS ONE.

Reviewer #2: Overall, the introduction provides a comprehensive overview of diabetes, its mechanisms, and potential treatments. However, there are some points that require further attention to ensure clarity, precision, and context. Here's a review of the introduction with suggestions for improvement:

1. Clarity and Precision:

• In the sentence "Diabetes increases the apoptosis rate...", the direct causation between diabetes and apoptosis should be substantiated or rephrased to avoid overstatement.

• In the last sentence, you mentioned "treatment with kerosene." It's a bit surprising because kerosene is generally known as a fuel. If there is a specific compound within kerosene that has therapeutic effects, it should be specified.

2. Citations:

• Ensure that the cited references correspond to the information given, and that there is no repetition in citations. For example, '(9)' is cited twice with different information.

3. Structural Flow:

• Consider reordering the details on the mechanics of DM, antioxidants, and ROS for better flow.

• The section about Crocin and crustin feels a bit out of place. It may be helpful to introduce why these compounds are relevant to the study, perhaps with a transition sentence.

4. Grammar and Wording:

• "...as high blood sugar interferes with the metabolism of fats, carbohydrates, and proteins." – This can be rephrased for clarity.

• In "Crocin and crustin are the most critical carotenoids responsible for the color of saffron.", it's unclear what "most critical" means. Does it mean they are the most abundant or most essential for the color?

• The phrase "not yet exposed to DNA" is unclear. Do you mean an enzyme that hasn't interacted with DNA yet?

5. Scientific Accuracy:

• "Apoptosis, or programmed cell death, is a natural and active process during evolution after cells are exposed to cytotoxic agents." – Apoptosis is a cellular process that can occur for various reasons, not just exposure to cytotoxic agents. The mention of evolution here is also a bit confusing.

• "...oxidative stress induced by acute exercise can affect the erythrocytes of non-exercised rats" – If there's a distinction between "non-exercised rats" and "exercised rats", it should be clearer.

6. Relevance:

• The relationship between crocin, crustin, diabetes, and oxidative stress should be made more explicit since this seems to be the premise of the study.

• The last sentence provides an objective for the study, but the connection between forced treadmill exercise, kerosene, and diabetes should be clearer.

In summary, the introduction provides valuable information but can benefit from improved organization, clarity, and precision in presentation. Ensure the relevance of all the information to the main topic, and make connections between sections explicit for better comprehension.

Materials and Methods provides a comprehensive insight into the methodology employed for the study. However, I would like to make a few suggestions and corrections to enhance clarity and precision.

1. Clarity on Animal Preparation:

• Clearly specify how long after inducing diabetes the treatments (like exercise and crocin administration) started.

• In the section detailing the groups, it's mentioned that a group was treated with kerosene, but there's no mention of kerosene in the subsequent details. Ensure there's consistency in the narrative.

2. Consistency with References:

• Ensure that all the references are consistently numbered and are in sequential order. For instance, there's a jump from (10) to (11) without any (11) mentioned.

3. Technical Corrections:

• Ensure you maintain consistency in your units and representations. For instance, the speed “10/55-83 m/min” is not clear. This probably needs correction.

• In the sentence, "blood glucose levels were measured by an EASY GLOCO device once a day at a 50 mg/kg dose by gavage," it's not clear what the "50 mg/kg dose by gavage" refers to. Was this the frequency of measuring glucose levels or related to the administration of streptozotocin or crocin?

4. Biochemical Analysis:

• When you mention “based on our previous study”, it’s crucial to provide a reference for that particular study so readers can trace back if needed.

5. Quantitative real-time polymerase chain reaction (RT-qPCR):

• It would be beneficial if you could provide a bit more detail on the thermal cycling conditions (denaturation, annealing, and extension temperatures and times).

6. Statistical Analysis:

• For a more robust statistical inference, mention if any post-hoc test (e.g., Tukey, Bonferroni) was used after the ANOVA.

7. General Writing:

• Some sentences are overly complex and might benefit from being split or restructured for clarity.

• Ensure that abbreviations are consistently used after being introduced the first time.

8. Histopathological evaluation:

• When you mention the rat heart tissue sections were "thoroughly scrutinized by a light microscope (Nikon) by two independent judges in a blindfold condition," consider using "blinded condition" or "double-blinded method" instead of "blindfold condition" for clarity.

Remember, the Materials and Methods section should offer enough detail so that another researcher could replicate the study. Ensure every step, condition, and protocol is described comprehensively.

Result

Histology figure in your results section lacks resolution, this can be a significant concern. Clear, high-resolution images are essential for readers and peer reviewers to make accurate assessments of your findings. Here's what you can consider doing:

1. Re-scanning or Re-imaging:

• If you still have access to the original histological slides, consider re-scanning or re-imaging them using a high-quality scanner or microscope equipped with a camera. Ensure that you're using the highest possible settings to capture the best resolution.

2. File Format:

• Save the image in a lossless file format like TIFF or PNG. These formats preserve image quality better than JPEG, which can lose some data every time the image is saved.

3. Compression:

• Avoid compressing the image. Compression can reduce the image's quality and resolution.

4. Figure Annotation:

• If annotations (like labels, arrows, etc.) were added to the image, ensure that the annotations themselves are clear and sharp. Use software that allows vector-based annotations, which won't degrade upon resizing.

5. Image Enhancement:

• If the image appears a bit unclear due to issues like low contrast, consider using image editing software to adjust the contrast, brightness, or sharpness. However, be very cautious; any modifications should be ethical, and they shouldn't change the underlying data or mislead the viewer. Always mention any enhancements made in the figure caption or materials and methods section.

6. Consider Alternative Representation:

• If the tissue stain or preparation was not clear, and you're unable to capture a high-resolution image, you might want to consider an alternative method to represent the histological findings. For example, a schematic or illustration might be used to complement the original figure. However, the original, unaltered image should always be provided.

7. Figure Caption:

• Ensure the figure caption provides sufficient details about the image. Mention the magnification, stain used, and any other pertinent details.

8. File Size Limitations:

• Some journals have limits on the file size of images. If your high-resolution image exceeds the journal's size limit, reach out to the editorial team. They might have provisions or suggestions for including large files.

Lastly, always double-check how the image looks in the manuscript, both on screen and printed. This will give you an idea of the clarity and quality that your readers will experience.

The discussion provided details on the complex relationship between diabetes mellitus (DM) and the subsequent complications it causes, particularly in the cardiovascular system. The analysis also focuses on the therapeutic potential of crocin, an antioxidant, combined with the benefits of regular exercise in mitigating these complications.

Key Points from the Discussion:

1. Introduction to Diabetes: DM is a metabolic disorder that leads to hyperglycemia, with the exact cause still uncertain but influenced by various factors like viral infections, autoimmune reactions, and environmental exposures.

2. Complications of Diabetes: Despite controlling glucose with insulin, patients with diabetes often face complications in various organs, notably the cardiovascular system. Oxidative stress plays a significant role in these complications.

3. Exercise and Oxidative Stress: Moderate physical activity has dual effects: while intense physical activity increases oxidative stress, regular and moderate-intensity exercise strengthens the body's antioxidant defense system.

4. Streptozotocin (STZ) & Diabetes Model: STZ is used experimentally to induce diabetes in animals, allowing for the study of potential therapeutic strategies.

5. Efficacy of Crocin & Exercise: In this study, both crocin and treadmill exercise demonstrated potential in controlling hyperglycemia and improving antioxidant defenses.

6. Mechanisms Behind the Benefits: The benefits seen might be due to flavonoids present in crocin that protect pancreatic beta cells and stimulate insulin secretion. Additionally, crocin, through its antioxidant properties, can prevent cell death, or apoptosis.

7. Lipid Profile Alterations: Diabetes alters lipid metabolism, resulting in an unfavorable lipid profile that contributes to cardiovascular complications. Crocin and exercise seem to improve this profile.

8. Apoptosis & Cardiac Tissue: Oxidative stress causes apoptosis in heart tissues. Bax and Bcl-2 are key proteins that decide the fate of a cell – whether it will undergo apoptosis or not. The balance between these two proteins (Bax/Bcl-2 ratio) can determine tissue outcomes. The study showed that crocin and regular exercise can favorably alter this balance, suggesting less tissue damage.

9. Conclusions:

• Diabetes exacerbates oxidative stress leading to damage in the heart tissue of rats.

• Oxidative stress markers like MDA increased, while antioxidant enzymes like SOD and GPx decreased in diabetic conditions.

• There was a notable increase in heart tissue damage in diabetic groups.

• Combining crocin and regular exercise offered protection against the detrimental effects of diabetes on the heart tissue of rats.

Recommendations for Future Research: To further validate these findings, it would be beneficial for future studies to:

1. Include larger sample sizes and diverse animal models.

2. Investigate the long-term effects of crocin and exercise.

3. Explore the potential benefits in human subjects.

4. Look into the potential synergistic effects of combining other antioxidants or therapeutic agents with crocin and exercise.

The study provides valuable insights into possible therapeutic strategies for managing diabetic complications. Integrating natural antioxidants like crocin with physical exercise offers a promising approach. However, translating these findings from animal models to human populations requires more extensive studies.

7. PLOS authors have the option to publish the peer review history of their article (what does this mean?). If published, this will include your full peer review and any attached files.

Reviewer #1: **Yes: **Angel Guerra-Moreno

Reviewer #2: No

---

## [Author Response · Author response to Decision Letter 1]

27 Sep 2023

Dear Editor-in-Chief;

Ref: Submission ID PONE-D-23-00007R1

 We thank the editor and reviewers for the time taken to conduct their reviews and the thoughtful feedback they provided. We have actioned/responded to the editor and reviewer feedback below and in the revised manuscript using the yellow background function. The authors have carefully considered the comments and tried our best to address every one of them.

In conclusion, we are pleased to see considerable support for our study and believe the paper is improved due to these changes. We believe the manuscript will interest your readership and audience and look forward to hearing from you in due course regarding this revision. 

Yours sincerely;

Journal Requirements:

Reviewers' comments:

Reviewer's Responses to Questions

Comments to the Author

1. If the authors have adequately addressed your comments raised in a previous round of review and you feel that this manuscript is now acceptable for publication, you may indicate that here to bypass the “Comments to the Author” section, enter your conflict of interest statement in the “Confidential to Editor” section, and submit your "Accept" recommendation.

Reviewer #1: (No Response)

Reviewer #2: (No Response)

2. Is the manuscript technically sound, and do the data support the conclusions?

Reviewer #1: Yes

Reviewer #2: Yes

3. Has the statistical analysis been performed appropriately and rigorously?

Reviewer #1: Yes

Reviewer #2: Yes

4. Have the authors made all data underlying the findings in their manuscript fully available?

Reviewer #1: No

Reviewer #2: No

5. Is the manuscript presented in an intelligible fashion and written in standard English?

Reviewer #1: No

Reviewer #2: Yes

6. Review Comments to the Author

AUTHOR RESPONSE:

Reviewer #1: Dear Dr. Hashemkandi,

I appreciate the paper and results you have presented. Although they are not entirely original, most of these insights have been described in previous publications by other research groups. However, the data provided here support and complement the previous findings. I particularly commend the simplicity and focus of the project, which was conducted entirely in vivo. The study contributes to our understanding of cardiovascular defects associated with diabetes and their contribution to the lethality in DM. It also explores new possibilities for patients to mitigate the impact of reactive oxygen species (ROS) on cardiac tissue, thereby increasing the potential to reduce risks. The authors should include references of important previous works in these areas. For instance, a study from the same center demonstrated beneficial effects of exercise on cardiovascular damage in diabetic rats (Naderi et al, 2015 Adv Pharm Bull). Additionally, previous studies have shown crocin, an antioxidant agent presents in saffron, reduces the impact of ROS in cardiovascular tissue in diabetics (Farshid et al, 2016, Avicenna J Phytomed). The findings of these papers should be discussed on the manuscript. Furthermore, it is essential to improve the English usage, grammar and vocabulary especially in the abstract. I agree that revised version is improved respect to the first one. Another general concern I have is regarding how the rat groups are described along the text. To enhance readability, I suggest using “diabetes group”, “diabetes treated group (exercise/crocin)” and “control/healthy group”, instead of “diabetes control group”. I also concur previous reviewer’s suggestion to provide access to the raw data. I highly recommend that the authors make it publicly available. Here some additional points that I believe should be addressed:

AUTHOR RESPONSE

Thank you for your comments. We have gone through your comments carefully and tried our best to address them one by one. We hope the manuscript has been improved accordingly.

1. I would like to know the exact number of rats in each of the 8 groups detailed in “Animal preparation and study design” section of Material and Methods.

Author's response: 

Thank you for your comments. This study used 64 rats in 8 groups (n=8).

2. In table 3, 4, 5 and 6 please indicate when the samples for the measurements were collected.

Author's response: 

Thank you for your comments. All measurements were done after the exercise and use of crocin.

3. For figures 1, 2 and 3, as well as tables 2 and 3, clarify the representation of

numbers/bars (average, median, …) and the type of error bars used (SEM, SE,..).

Author's response: 

Thanks for your feedback. We clarified central tendency and dispersion measures in Tables 2 and 3. It also described the error bar type SD standard Deviation in Figures 1, 2, and 3. Revisions were highlighted in the figures and tables.

4. Include reference to previous studies that have been assessed the effects of crocin and exercise on generation of cardiovascular damage in diabetes models.

Author's response: 

Thanks for your feedback. Your suggested studies were added and highlighted.

5. Consider improving the visual presentation of tables by using graphs, particularly for tables 5 and 6.

Author's response:

Thanks for your feedback. We clarified central tendency and dispersion measures in Tables 5 and 6. We observed that to show all data better, we must use table format.

6. In the histological studies, I highly recommend implementing a method to quantify the extent of histological damage. This way you can strengthen the analysis and provide more robust assessment of the observed histopathological changes. This also will enhance the objectivity and reliability of the findings and will complement the microscope images.

Author's response:

Thank you for your valuable feedback and suggestions regarding the histological studies in our manuscript. We appreciate your insightful recommendation to implement a method for quantifying the extent of histological damage, which could strengthen our analysis and provide a more robust assessment of the observed histopathological changes. We acknowledge the importance of quantification in histological studies to enhance objectivity and reliability while complementing the microscope images. However, we regret to inform you that, due to limited resources and current institutional constraints, we cannot access specialized software or tools that can effectively analyze H&E staining results for quantification purposes. Despite this limitation, we have taken several steps to address the issue and provide the most thorough analysis possible within our constraints. We have ensured that the histopathological assessments were conducted by experienced pathologists who followed established guidelines and employed a consistent grading system for the qualitative evaluation. While this may not provide quantitative data, it offers valuable qualitative insights into the observed changes. We have also included detailed descriptions and figures to illustrate the histological findings, allowing readers to assess the extent of damage visually.

Moreover, in the discussion section of the manuscript, we have emphasized the qualitative aspects of the histological results and their clinical implications. We understand that quantification would be a valuable addition to our study and appreciate your understanding of our limitations. We hope the comprehensive qualitative analysis we have provided and the other rigorous aspects of our research can still contribute meaningfully to the field.

Once again, we thank you for your constructive feedback, and we remain open to any further suggestions or comments you may have to improve the quality and impact of our manuscript.

7. It would be beneficial to include comments in the text regarding the recovery of kidney function after exercise and crocin treatment, supported by the levels of creatinine and urea. Additionally, I have some suggestions to enhance the comprehensiveness of the paper and introduce further novelty by incorporating complementary readouts to the experiments:

Author's response: 

Thanks for your feedback. Your suggested studies were added and highlighted. About the suggested evaluation tests. First, we would like to thank you for your suggestion. But, since our study was completed at least 1 year ago, we cannot do your mentioned test.

1. In addition to measuring of MDA, SOD and GPx to asses oxidative stress, that is an indirect way, consider including another assay, there are a bunch of kits that help to confirm an increase of the oxidative stress, I will recommend to use one to check the oxidation levels of the heart proteins in the 8 groups of rats.

Author's response: 

First, we would like to thank you for your suggestion. But, since our study was completed at least 1 year ago, we cannot do your mentioned test.

2. To complement the histological approach and confirm apoptosis in a direct way, consider conducting a functional assay, a very easy one will be to measuring annexin V levels by flow cytometry in rat heart cells. Additionally, to investigate apoptosis induction, which is determined by the balance of pro- and anti- apoptotic regulators, consider measuring the levels of other apoptotic regulators by qPCR (Bcl x, MCL1, BAD, BIM,…) this way results would be more robust. Alternatively, BH3 profiling offers a novel and interesting approach to test apoptosis. In conclusion, I find the work very promising, and with the necessary improvements, I would recommend that the editor accept it for publication in PLOS ONE.

Author's response: 

First, we would like to thank you for your suggestion. But, since our study was completed at least 1 year ago, we cannot do your mentioned test.

Reviewer #2: Overall, the introduction provides a comprehensive overview of diabetes, its mechanisms, and potential treatments. However, there are some points that require further attention to ensure clarity, precision, and context. Here's a review of the introduction with suggestions for improvement:

AUTHOR RESPONSE

Thank you for your comments. We have gone through your comments carefully and tried our best to address them one by one. We hope the manuscript has been improved accordingly.

1. Clarity and Precision:

• In the sentence "Diabetes increases the apoptosis rate...", the direct causation between diabetes and apoptosis should be substantiated or rephrased to avoid overstatement.

AUTHOR RESPONSE

Thanks for your feedback. Your suggested sentence was rewritten and highlighted.

Diabetes increases the apoptosis rate (Pancreatic β-cell apoptosis is also a pathological feature common to Type 1 diabetes mellitus (T1DM) and T2DM. In T2DM, insulin resistance with visceral obesity leads to a glucose toxicity effect, which accelerates β-cell death by apoptosis) in heart cells and disrupts cardiac function

• In the last sentence, you mentioned "treatment with kerosene." It's a bit surprising because kerosene is generally known as a fuel. If there is a specific compound within kerosene that has therapeutic effects, it should be specified.

AUTHOR RESPONSE

Thanks for your feedback. Your suggested sentence was rewritten and highlighted.

Accordingly, this study aimed to determine the effect of forced treadmill exercise and concomitant treatment with crocin on heart tissue damage and oxidative stress in diabetic rats.

2. Citations:

• Ensure that the cited references correspond to the information given, and that there is no repetition in citations. For example, '(9)' is cited twice with different information.

AUTHOR RESPONSE

Thanks for your attention. Checked and corrected.

3. Structural Flow:

• Consider reordering the details on the mechanics of DM, antioxidants, and ROS for better flow.

AUTHOR RESPONSE

Thanks for your attention. Checked and corrected.

• The section about Crocin and crustin feels a bit out of place. It may be helpful to introduce why these compounds are relevant to the study, perhaps with a transition sentence.

AUTHOR RESPONSE

Thanks for your attention. It was checked, rewritten, and highlighted in the manuscript. 

Crocin and crustin are the predominant carotenoids that play a pivotal role in determining the coloration of saffron. Crocin undergoes metabolic processes within the human body, forming Crocin, which exhibits numerous medicinal qualities.

4. Grammar and Wording:

• "...as high blood sugar interferes with the metabolism of fats, carbohydrates, and proteins." – This can be rephrased for clarity.

AUTHOR RESPONSE

Thanks for your attention. It was checked, rewritten, and highlighted in the manuscript. 

Diabetes mellitus is a syndrome characterized by lipid, carbohydrate, and protein metabolism disruption due to elevated blood sugar levels. Consequently, it has the potential to heighten the susceptibility to vascular disease.

• In "Crocin and crustin are the most critical carotenoids responsible for the color of saffron.", it's unclear what "most critical" means. Does it mean they are the most abundant or most essential for the color?

AUTHOR RESPONSE

Thanks for your attention. It was checked, rewritten, and highlighted in the manuscript.

• The phrase "not yet exposed to DNA" is unclear. Do you mean an enzyme that hasn't interacted with DNA yet?

AUTHOR RESPONSE

Thanks for your attention. It was checked, rewritten, and highlighted in the manuscript.

About the suppressive impact of crocin on tumorigenesis, a plausible hypothesis may be formulated suggesting that crocin functions as a stimulator for a DNA-repair enzyme that has not yet encountered DNA.

5. Scientific Accuracy:

• "Apoptosis, or programmed cell death, is a natural and active process during evolution after cells are exposed to cytotoxic agents." – Apoptosis is a cellular process that can occur for various reasons, not just exposure to cytotoxic agents. The mention of evolution here is also a bit confusing.

AUTHOR RESPONSE

Thanks for your attention. It was checked, rewritten, and highlighted in the manuscript.

Apoptosis, also known as programmed cell death, is an inherent and dynamic biological process that occurs in response to exposure of cells to cytotoxic substances, as observed throughout evolution. Apoptosis is characterized by several prominent aspects, namely cellular shrinkage, membrane impairment, chromatin condensation, and fragmentation of DNA. Multiple proteins have a role in the regulation of apoptosis. The Bcl-2 protein family encompasses a group of proteins that exert regulatory control over apoptosis, playing pivotal roles in inhibiting and promoting this cellular process. While the Bcl-2 protein functions as a suppressor, the Bax protein serves as a promoter of apoptosis.

• "...oxidative stress induced by acute exercise can affect the erythrocytes of non-exercised rats" – If there's a distinction between "non-exercised rats" and "exercised rats", it should be clearer.

AUTHOR RESPONSE

Thanks for your attention. It was checked, rewritten, and highlighted in the manuscript.

It has been shown that oxidative stress induced by acute exercise can affect the erythrocytes of non-exercised rats" and "exercised rats, while it does not significantly affect the erythrocytes of exercised rats.

6. Relevance:

• The relationship between crocin, crustin, diabetes, and oxidative stress should be made more explicit since this seems to be the premise of the study.

AUTHOR RESPONSE

Thanks for your attention. It was entirely considered in the manuscript.

• The last sentence provides an objective for the study, but the connection between forced treadmill exercise, kerosene, and diabetes should be clearer.

AUTHOR RESPONSE

Thanks for your attention. It was checked, rewritten, and highlighted in the manuscript.

In summary, the introduction provides valuable information but can benefit from improved organization, clarity, and precision in presentation. Ensure the relevance of all the information to the main topic, and make connections between sections explicit for better comprehension.

Materials and Methods provides a comprehensive insight into the methodology employed for the study. However, I would like to make a few suggestions and corrections to enhance clarity and precision.

AUTHOR RESPONSE

Thank you for your comments. We have gone through your comments carefully and tried our best to address them one by one. We hope the manuscript has been improved accordingly.

1. Clarity on Animal Preparation:

• Clearly specify how long after inducing diabetes the treatments (like exercise and crocin administration) started.

AUTHOR RESPONSE

Thank you for your comments. One week after the STZ injection, rats exhibiting hyperglycemia were considered type 1 diabetic and subjected to subsequent experiments.

• In the section detailing the groups, it's mentioned that a group was treated with kerosene, but there's no mention of kerosene in the subsequent details. Ensure there's consistency in the narrative.

AUTHOR RESPONSE

Thank you for your comments. Our goal was crocin, so the corrected word was replaced.

2. Consistency with References:

• Ensure that all the references are consistently numbered and are in sequential order. For instance, there's a jump from (10) to (11) without any (11) mentioned.

AUTHOR RESPONSE

Thank you for your comments. All of the references were double-checked.

3. Technical Corrections:

• Ensure you maintain consistency in your units and representations. For instance, the speed “10/55-83 m/min” is not clear. This probably needs correction.

AUTHOR RESPONSE

Thank you for your comments. The required part was corrected.

• In the sentence, "blood glucose levels were measured by an EASY GLOCO device once a day at a 50 mg/kg dose by gavage," it's not clear what the "50 mg/kg dose by gavage" refers to. Was this the frequency of measuring glucose levels or related to the administration of streptozotocin or crocin?

AUTHOR RESPONSE

Thanks for your attention. It was checked, rewritten, and highlighted in the manuscript.

After 72 hours, blood glucose levels were measured by an EASY GLOCO once daily. Then, crocin at a 50 mg/kg dose by gavage to the rats.

4. Biochemical Analysis:

• When you mention “based on our previous study”, it’s crucial to provide a reference for that particular study so readers can trace back if needed.

AUTHOR RESPONSE

Thanks for your attention. It was checked, and required reference was added.

5. Quantitative real-time polymerase chain reaction (RT-qPCR):

• It would be beneficial if you could provide a bit more detail on the thermal cycling conditions (denaturation, annealing, and extension temperatures and times).

AUTHOR RESPONSE

Thanks for your attention. We provided details of denaturation, annealing and temperature and times for RT-aPCR in text and highlighted by yellow background under (Methods, Quantitative real-time polymerase chain reaction (RT-qPCR, line …).

6. Statistical Analysis:

• For a more robust statistical inference, mention if any post-hoc test (e.g., Tukey, Bonferroni) was used after the ANOVA.

AUTHOR RESPONSE

Thanks for your attention. SPSS software (version 21.0, Chicago, IL, USA) was used for data analysis. One-way analysis of variance (ANOVA) followed by a multiple two-tailed T-test was used to study significant differences among the collected data. P <0.05 was considered statistically significant.

7. General Writing:

• Some sentences are overly complex and might benefit from being split or restructured for clarity.

AUTHOR RESPONSE

Thanks for your attention. General Writing of the manuscript was double-checked.

• Ensure that abbreviations are consistently used after being introduced the first time.

AUTHOR RESPONSE

Thanks for your attention. All of them were double-checked.

8. Histopathological evaluation:

• When you mention the rat heart tissue sections were "thoroughly scrutinized by a light microscope (Nikon) by two independent judges in a blindfold condition," consider using "blinded condition" or "double-blinded method" instead of "blindfold condition" for clarity.

Remember, the Materials and Methods section should offer enough detail so that another researcher could replicate the study. Ensure every step, condition, and protocol is described comprehensively.

AUTHOR RESPONSE

Thanks for your attention. It was corrected and highlighted in the manuscript. 

Material methods were rechecked.

Result

Histology figure in your results section lacks resolution, this can be a significant concern. Clear, high-resolution images are essential for readers and peer reviewers to make accurate assessments of your findings. Here's what you can consider doing:

1. Re-scanning or Re-imaging:

• If you still have access to the original histological slides, consider re-scanning or re-imaging them using a high-quality scanner or microscope equipped with a camera. Ensure that you're using the highest possible settings to capture the best resolution.

Authors response: 

Thanks for your comment. We quantified the level and/or value of the histological damage based on the study groups as shown in Figure 4 (microscope images).

2. File Format:

• Save the image in a lossless file format like TIFF or PNG. These formats preserve image quality better than JPEG, which can lose some data every time the image is saved.

Authors response: 

Thanks for your comment. All of the figures were saved in TIFF format.

3. Compression:

• Avoid compressing the image. Compression can reduce the image's quality and resolution.

Authors response: 

Thanks for your considerable comment.

4. Figure Annotation:

• If annotations (like labels, arrows, etc.) were added to the image, ensure that the annotations themselves are clear and sharp. Use software that allows vector-based annotations, which won't degrade upon resizing.

Authors response: 

Thanks for your considerable comment.

5. Image Enhancement:

• If the image appears a bit unclear due to issues like low contrast, consider using image editing software to adjust the contrast, brightness, or sharpness. However, be very cautious; any modifications should be ethical, and they shouldn't change the underlying data or mislead the viewer. Always mention any enhancements made in the figure caption or materials and methods section.

Authors response: 

Thanks for your considerable comment.

6. Consider Alternative Representation:

• If the tissue stain or preparation was not clear, and you're unable to capture a high-resolution image, you might want to consider an alternative method to represent the histological findings. For example, a schematic or illustration might be used to complement the original figure. However, the original, unaltered image should always be provided.

Authors response: 

Thanks for your considerable comment.

7. Figure Caption:

• Ensure the figure caption provides sufficient details about the image. Mention the magnification, stain used, and any other pertinent details.

Authors response: 

Thanks for your considerable comment. It was considered.

8. File Size Limitations:

• Some journals have limits on the file size of images. If your high-resolution image exceeds the journal's size limit, reach out to the editorial team. They might have provisions or suggestions for including large files.

Authors response: 

Thanks for your considerable comment. It was considered.

Lastly, always double-check how the image looks in the manuscript, both on screen and printed. This will give you an idea of the clarity and quality that your readers will experience.

AUTHOR RESPONSE

Thank you for your comments. We have gone through your comments carefully and tried our best to address them one by one. We hope the manuscript has been improved accordingly.

---

## [Editor Report · Decision Letter 2]

29 Sep 2023

Effect of crocin and treadmill exercise on oxidative stress and heart damage in diabetic rats

PONE-D-23-00007R2

Dear Dr. hashemkandi,

We’re pleased to inform you that your manuscript has been judged scientifically suitable for publication and will be formally accepted for publication once it meets all outstanding technical requirements.

Kind regards,

Wajdy Jum’ah Al-Awaida, Ph.D

Academic Editor

PLOS ONE
---

## [Editor Report · Acceptance letter]

23 Oct 2023

PONE-D-23-00007R2 

Effect of crocin and treadmill exercise on oxidative stress and heart damage in diabetic rats 

Dear Dr. Hashemkandi Asadi:

I'm pleased to inform you that your manuscript has been deemed suitable for publication in PLOS ONE. Congratulations! Your manuscript is now with our production department. 

Kind regards, 

on behalf of

Prof. Wajdy Jum’ah Al-Awaida 

Academic Editor

PLOS ONE